# Shared Paths to Well-Being: The Impact of Group Therapy

**DOI:** 10.3390/bs15010057

**Published:** 2025-01-10

**Authors:** Elena Renée Sequeira-Nazaré, Bernhard Schmitz

**Affiliations:** Department of Human Sciences, Technische Universität Darmstadt, 64289 Darmstadt, Germany; schmitz@psychologie.tu-darmstadt.de

**Keywords:** psychotherapy, depression, art of living, life satisfaction, self-efficacy, flourishing, mental health

## Abstract

This study explored the impact of an art of living intervention within group psychotherapy for depression, focusing on constructs like life satisfaction, self-efficacy, and depression. Mental illness prevalence often exceeds available treatment options, particularly in Germany, where group psychotherapy is a viable alternative. While less researched, group therapy effectively improves well-being, especially through interpersonal exchange. Meta-analyses confirm cognitive behavioral group therapy’s effectiveness against depression, encouraging further investigation. This study employed a two-factor experimental design with randomized group allocation. The control group (CG) participated in weekly 50 min sessions for four weeks, while the experimental group (EG) received identical therapy plus reflective life-stimulating questions. Measures of depression, art of living, life satisfaction, and self-efficacy were taken before, after, and three months post-intervention. Among 107 participants, 52 were in the EG and 55 were in the CG. The results showed a significant 24% reduction in depression scores in the experimental group, a significant 16% increase in the art of living and a significant 19% increase in life satisfaction, while the CG showed no significant changes. Self-efficacy did not significantly improve in the EG. Follow-up data indicated sustained improvements in depression and art of living for the EG. The limitations of this study include a limited scope, practical constraints, randomization challenges and confounding variables, which are typical for experimental studies. These findings highlight the intervention’s potential, suggesting future research focusing on long-term effects, personality factors and disorder-specific applications.

## 1. Introduction

The prevalence of mental illnesses exceeds the scope of treatment services. Group psychotherapy could counteract this imbalance.

In 2015, around a quarter of men and a third of women in Germany suffered from a mental disorder ([12]). Studies show the effectiveness and long-term impact of psychotherapy, which underlines the increasing importance of psychotherapeutic support and research ([22]; [9]; [16]).

Inpatient psychotherapy includes various approaches such as individual and group therapy and creative methods. A study by [31] ([31]) shows the effectiveness of psychotherapy, with a significant reduction in depression scores by the time of discharge.

Despite its frequent integration into comprehensive treatment concepts, research on group therapy receives little scientific attention compared to individual therapy ([34]).

Numerous studies have shown that group therapy is equally as effective as individual therapy, particularly in the treatment of depression ([4]; [10]; [29]). [6] ([6]) conducted a meta-analysis on the effectiveness of individual and group therapy for depression. They analyzed the short- and long-term effectiveness in 15 studies. Individual therapy showed superior efficacy in the short term, with a small effect size of d = 0.20. In the long term, however, no significant differences were found between the two forms of therapy.

In a semi-experimental study involving 50 participants divided into experimental and control groups, those undergoing group therapy showed marked improvements compared to the control group. The results showed that cognitive behavioral group therapy is highly effective for women with breast cancer, significantly reducing depression and anxiety while improving pain-coping strategies. The findings highlight the therapy’s potential as a beneficial intervention for managing psychological and pain-related challenges ([20]).

One study ([23]) analyzed the effectiveness of group cognitive behavioral therapy (CBT) for patients with chronic illnesses and depression. The findings indicate that group CBT has comparable, if not superior, effects to individual therapy, particularly in reducing depressive symptoms and improving overall well-being.

A meta-analysis ([32]) focused on cognitive behavioral therapy in group settings. It showed that group therapy is very effective in the treatment of depression and that group therapies have a positive effect on social support, self-efficacy and depressive symptoms.

The particular therapeutic relevance of the group effect factors results from the fact that they are not present as mechanisms of therapeutic change in individual therapy, at least not to the same intensity ([11]). In contrast to individual therapy, there are other interpersonal principles in groups that can be used to optimize interaction problems. The empirical literature indicates that the conscious promotion of individual factors by the group leader can contribute to positive changes in patients quite independently of the patient population and of specific effective factors, such as obtaining and giving feedback, support, altruism, model learning or role-playing, which are highly relevant in behavioral therapy groups ([11]).

On an emotional level, a patient finds numerous opportunities for affect projection in various group members. Compared to indirect methods, group discussions have a deeper analytical effect, while they are less intensive than individual therapy.

Despite numerous positive results, however, the drop-out rate in group therapy appears to be higher than in individual therapy ([4]). One study ([18]) reports a drop-out rate of between 17% and 55.5% of patients in their review. On average, a third of patients drop out during group therapy ([30]). However, the high complexity of group processes makes an empirical scientific approach difficult.

According to [15] ([15]), the overall cost ratio is calculated as 13:1 for individual vs. group therapy. This describes the ratio between the total costs incurred for group or individual therapy and the benefits they create. This means that the costs for individual therapy are 13 times higher than for group therapy.

For most patients, a rapid restoration of positive emotions such as joie de vivre and motivation is more important than simply alleviating depressive moods ([1]). Doctors, on the other hand, place more emphasis on reducing negative emotions ([7]). Studies show that increased well-being reduces the risk of mental illness and alleviates symptoms ([33]).

The art of living describes a conscious and reflective approach to the self and to life according to [25] ([25]). In psychotherapy, the art of living is important because well-being is more than just the absence of illness. It encompasses the self, the body, the soul, the environment and the mind, with self-reflection playing a central role. This enables us to think about our own actions, evaluate them and develop further ([5]). Studies show that self-reflection is associated with increased well-being, especially when it results in alternative courses of action for the future ([19]).

In a study by [27] ([27]), the author explores the relationships between the art of living and positive psychology constructs such as strengths and virtues. The study focuses on how art of living practices intersect with character strengths and virtues to promote a fulfilling and meaningful life. The study validates that the art of living aligns closely with key constructs of positive psychology, emphasizing virtues like wisdom, courage, and self-regulation. Strengths and virtues play a mediating role in the art of living, directly contributing to an individual’s overall happiness and life satisfaction. The findings suggest that fostering specific art of living practices, such as mindfulness and self-reflection, enhances strengths, which in turn promotes well-being. This work underlines the integration of philosophical principles and positive psychology, emphasizing that cultivating strengths through art-of-living strategies can lead to a happier, more virtuous life.

Another study ([28]) investigates the influence of a life skills intervention within psychotherapy for depressed patients on life skills, flourishing, SWLS and depression. In the first experimental group (EG1), participants received a 50 min psychotherapy session every week for four weeks. The second experimental group (EG2) took part in the same sessions but also answered daily reflection questions on the art of living, which were recorded in a diary. The control group (CG) received neither therapy nor reflection questions. The two groups were allocated in advance by the clinic staff. The allocation of which of the two groups would be EG1 or EG2 and which would be CG was randomized. Before and after the interventions and three months later, life skills, self-efficacy, the degree of depression and life satisfaction were recorded and compared. In both experimental groups, there was a significant reduction in depression symptoms over the four-week period but no significant difference between the groups. In EG2, there was a significant increase in the art of living, while in EG1, there was no significant change. No significant changes were found with regard to the Flourishing Scale. EG2 also showed a significant increase in life satisfaction, while no significant changes were discernible in EG1.

Based on this study, this paper examines the influence of an art of living intervention within group psychotherapy for depressed patients on the art of living, flourishing, SWLS and depression.

## 2. Materials and Methods

Informed written consent was obtained from the participants through the questionnaire, and the Ethics Committee of TU Darmstadt waived the requirement for additional consent. The study did not include any minors. It followed a two-factorial experimental design, involving group comparisons between two groups and three measurement repetitions. The duration of the study was 10 months.

The groups were pre-assigned by the clinic prior to the study, with the primary criterion being the placement of alcohol-addicted patients into one group. The experimenter, who was blinded to the group assignments, then randomly designated the groups as either experimental or control. Participants in the control group (CG) attended one 50 min group session per week for four weeks. The experimental group (EG) received the same psychotherapy as the CG, with the addition of life-stimulating reflection questions, which each participant answered weekly. The clinic determined the group composition, and the assignment of participants to either the CG or EG was randomized. Both groups were part of ongoing therapy sessions, a format that allows for new participants to join the group continuously while others leave after a certain period. On average, there was about one change per week. Despite this, the stable structure of the program ensured that all participants received the same interventions, allowing everyone to benefit equally, regardless of when they joined the group. Regarding repeated measures, assessments of art of living, depression levels, life satisfaction and flourishing were conducted before and after the interventions, as well as three months following the interventions. The exclusion criteria for all groups included being underage and having a BDIV score below the cut-off value of 35.

The interventions represent the independent variable, the questionnaires (AOL = art of living; BDI-V = depression; SWE = self-efficacy; FS = Flourishing Scale) and the dependent variable.

The BDI (Beck Depression Inventory) is an early self-report scale for assessing depressive symptoms, consisting of 20 items measured on a five-point Likert scale, with a Cronbach’s alpha of 0.95 ([24]). The art of living questionnaire, a short form of a longer validated version, includes 35 items across 11 subscales, such as self-determined lifestyle, self-knowledge, and social contacts, measured on a six-point Likert scale, with a Cronbach’s alpha of 0.92 ([26]). The Flourishing Scale, an 8-item measure of psychological well-being in areas like relationships and self-esteem, has a Cronbach’s alpha of 0.87 ([8]). The Satisfaction with Life Scale, a five-item measure of life satisfaction with both affective and cognitive–evaluative components, uses a 7-point Likert scale and has a Cronbach’s alpha of 0.85 ([17]; [21]).

The sample consisted of voluntary patients from the Varisano Clinic who suffered from at least mild depression.

The total sample consisted of 107 voluntary participants, 70% of whom were female, and 28% were male. There were two gender-neutral individuals. A total of 55 people were in the CG, and 52 were in the EG. The average age of all patients was M = 43.68 years, with SD = 14.1 years. The groups differed in their gender composition (CG: 71% female; EC: 69% female).

The youngest patient was 20 years old, and the oldest was 78 years old.

### 2.1. Analysis and Description of the Experimental Intervention

For the control group (CG), psychotherapy in the form of group therapy served as the intervention. The content of the group therapy included psychoeducation, behavior analysis, emotional regulation, and confrontation exercises ([14]). The experimental group (EG) also received psychotherapy as group therapy, but in addition, the participants worked on reflection questions designed by the experimenter, based on the work of [25] ([25]). These reflection questions were completed by all participants in the group and included the following:

Reflection on today: 1. What did I enjoy today (even if only in a very small way)? 2. What am I proud of today (even if only in a very small way)? 3. What am I grateful for today? 4. If I could relive the day, would I do anything differently? If so, what would that be? Reflection on tomorrow: 5 Assuming I have all the strength in the world tomorrow to do what I set out to do, what would that be? 6. How would it make me feel? 7. How would people close to me notice that I was feeling better?

The goal was for participants to answer these questions each week over a period of four weeks during group therapy and to engage in discussions with the other group members.

### 2.2. Hypotheses

**H1:** 
*The depression level of the EG and CG decreases significantly in the before–after comparison over the 4 weeks.*


**H2:** 
*Art of living, self-efficacy and life satisfaction increase significantly in EG1 and the CG over the 4 weeks.*


**H3:** 
*There are significant differences between the EG and the CG with regard to the change in the art of living, life satisfaction and self-efficacy over a period of 4 weeks, as well as with regard to the level of depression.*


**H4:** 
*In the 3-month post hoc measurement, there are significant differences in the stability of depression (BDI) and positive constructs (SWE, FS, LK) between the EG and the CG.*


### 2.3. Statistical Analysis

The trajectories of depression levels, self-efficacy, life skills and life satisfaction were analyzed in both groups (EG and CG) over three measurement time points using a two-factor analysis of variance with repeated measures (ANOVA). The Greenhouse–Geisser correction is reported in order to correct for sphericity violations.

## 3. Results

As expected, depression and the positive constructs of positive psychology correlate negatively with each other: depression and flourishing show a strong negative correlation (r = −0.630**), as do depression and self-efficacy (r = −0.471**) and depression and the art of living (r = −0.592**). This indicates that higher depression scores are associated with less experience of the positive constructs.

The positive constructs, on the other hand, correlate positively with each other: flourishing and self-efficacy show a strong positive correlation (r = 0.761**), flourishing and art of living correlate (r = 0.816**), and self-efficacy and art of living also show a positive correlation (r = 0.671**). These positive correlations reflect the expected mutual reinforcement between positive psychological aspects. Additional information can be found in the Appendix A.

### 3.1. Depression

There was no significant main effect across the groups, with F(1.54, 80.20) = 0.267, *p* < 0.708, and partial η^2^ = 0.005, indicating that there were no significant differences in depression levels across all the patients. The interaction between the groups (EG and KG) and the measurement times, on the other hand, proved to be significant, with F(1.542, 80.20) = 7.794, *p* = 0.002, and partial η^2^ = 0.130, which means that the changes in depression levels between the control group (KG) and the experimental group (EG) differed significantly from each other.

In detail, the pairwise comparisons showed that no significant changes occurred in the control group (CG) across the measurement time points. The mean value of the depression level remained almost unchanged between t0 (M = 60.77, SD = 14.14), t1 (M = 60.59, SD = 14.19) and t2 (M = 58.09, SD = 19.60). In the experimental group (EG), on the other hand, there was a significant reduction in the level of depression. A significant decrease was observed between t0 (M = 66.21, SD = 9.57) and t1 (M = 52.15, SD = 10.75) (*p* < 0.001), whereas the reduction between t1 and t2 (M = 50.12, SD = 11.14) was not significant (*p* = 0.459).

In Figure 1, the experimental group (red) begins therapy with higher average depression scores. These scores decrease significantly after four weeks, followed by a smaller reduction observed in the follow-up test. However, the control group (blue) starts with lower average depression scores, showing only a slight reduction over time. Similarly, the follow-up test scores exhibit only a minimal further decrease.

In summary, Hypothesis 1, which states that the level of depression in both groups decreases significantly in the before–after comparison after four weeks, was only confirmed for the experimental group (EG). In contrast, there were no significant changes in the control group (CG). Hypothesis 3 can therefore also be confirmed, namely that the two groups differ in terms of the reduction in depression, with a greater reduction in the experimental group. In order to address H4, which postulates that there are significant differences in the stability of depression scores between the experimental group (EG) and the control group (KG) in the 3-month post hoc measurement, the pairwise comparison between measurement time 1 (t0) and measurement time 3 (t2) was considered. In the experimental group (EG), the mean value decreased significantly at *p* < 0.001 from t0 (M = 66.21, SD = 9.57) to t2 (M = 50.12, SD = 11.14), while the slight decrease in the control group from t0 (M = 60.77, SD = 14.14) to t2 (M = 58.09, SD = 19.60) did not prove to be significant. Hypothesis 4 can be confirmed for the construct depression.

### 3.2. Self-Efficacy

The ANOVA revealed no significant main effect of measurement time points, with F(1.75, 96.26) = 0.093, *p* = 0.887, and partial η^2^ = 0.002, indicating that there were no significant changes in self-efficacy across all the patients. The interaction between the groups (EG and KG) and the measurement time points was also not significant, with F(1.75, 96.26) = 0.128, *p* = 0.854, and partial η^2^ = 0.002, meaning that the changes in self-efficacy did not differ significantly between the two groups. In detail, the pairwise comparisons showed that no significant changes occurred in the control group (CG) across the measurement points. The mean value of self-efficacy remained relatively stable between t0 (M = 13.67, SD = 5.04), t1 (M = 14.67, SD = 5.86) and t2 (M = 13.08, SD = 5.71). In the experimental group (EG), the increase in self-efficacy from t0 (M = 15.29, SD = 5.29) to t1 (M = 17.83, SD = 5.36) was not significant, and it also remained stable between t1 and t2 (M = 17.37, SD = 5.56). In Figure 2 the experimental group (red) begins with higher average self-efficacy scores compared to the control group and shows an increase after four weeks. However, these scores slightly decrease in the follow-up. The control group (blue) also experiences an increase in self-efficacy scores, though not as pronounced as the experimental group. In the follow-up, the control group’s scores decrease more than those of the experimental group. In summary, hypothesis 2, which states that self-efficacy increases over time in both groups, could not be confirmed, nor could the hypothesis of a difference over the 4 weeks between the groups (hypothesis 3). The differences between t0 and t3 were not significant in either group. As a result, there were no significant differences in the stability of self-efficacy between the experimental group (EG) and the control group (KG) in the 3-month follow-up measurement. Hypothesis 4 must therefore be rejected.

### 3.3. Art of Living

The ANOVA showed no significant main effect of measurement time (MSW), with F(1.995, 109.74) = 0.142, *p* = 0.867, and partial η^2^ = 0.003, indicating that there were no significant changes in the art of living across both groups over time. The interaction between the groups (EG and CG) and the measurement time, however, proved to be significant, with F(1.995, 109.74) = 7.012, *p* = 0.001, and partial η^2^ = 0.113, meaning that the changes in the art of living in the two groups differed significantly from each other. In the control group (CG), the art of living values remained relatively stable across the three measurement times. The mean was M = 3.15, with an SD = 0.73, at t0, and this remained almost unchanged at t1 (M = 3.15, SD = 0.89) and decreased slightly at t2 (M = 2.92, SD = 1.08). However, these changes were not significant. In the experimental group (EG), on the other hand, there was a significant increase in the art of living over time. The mean increased significantly from t0 (M = 3.04, SD = 0.71) to t1 (M = 3.48, SD = 0.60, *p* < 0.001) and then remained relatively stable (M = 3.51, SD = 0.72). In Figure 3, the experimental group (red) begins with lower average art of living scores, which increase over the four weeks but show only minimal improvement in the follow-up. In contrast, the control group (blue) starts with slightly higher average art of living scores than the experimental group, but these scores slightly decrease after four weeks and continue to decline slightly in the follow-up. In summary, hypothesis 2, which states that the art of living increases over the 4 weeks, was confirmed in the experimental group (EG). The art of living values increased significantly over time. In the control group (CG), however, no significant changes were seen, and the art of living scores remained largely unchanged, so hypothesis 3 can be confirmed; the experimental group showed a significantly higher increase than the control group. While the increase in the art of living scores in the experimental group was also significant in the 3-month post hoc measurement (*p* < 0.001), no significant difference was seen in the control group (see above). Hypothesis 4, which conjectured a significant difference in the art of living, can thus be confirmed in favor of the experimental group.

### 3.4. Life-Satisfaction/Flourishing

The analysis of variance revealed no significant main effect of time, with F(1.98, 107.03) = 2.638, *p* = 0.856, and η^2^ = 0.003, indicating that there were no significant changes in life satisfaction across all the patients. The interaction between the groups and the measurement times was also not significant, with F(1.98, 107.03) = 2.008, *p* = 0.139, and η^2^ = 0.036, indicating no significant differences in the changes in life satisfaction between the two groups. The pairwise comparisons showed no significant changes in the control group (CG). The mean value of life satisfaction remained relatively stable between t0 (M = 27.96, SD = 10.54), t1 (M = 28.29, SD = 9.79) and t2 (M = 25.21, SD = 11.15). However, in the experimental group (EG), there was a significant increase in life satisfaction from t0 (M = 24.21, SD = 8.41) to t1 (M = 28.91, SD = 7.53) at *p* < 0.001, with the values remaining stable at t2 (M = 28.15, SD = 7.91). In Figure 4 the experimental group (red) starts therapy with lower average flourishing scores, which increase over the course of the four weeks but decrease slightly in the follow-up. In contrast, the control group (blue) begins with higher average flourishing scores, showing only a slight increase after the four weeks, followed by a decline in the follow-up. In summary, hypothesis 2, which states that life satisfaction increases significantly in both groups, could only be confirmed for the experimental group (EG). No significant changes were seen in the control group (CG), so hypothesis 3 can be confirmed; the experimental group showed a significantly higher increase than the control group. The increase in life satisfaction in the experimental group was also significant in the 3-month post hoc measurement (*p* = 0.045), while no significant difference was found in the control group. Hypothesis 4, which assumed that there would be a significant difference in life satisfaction, was thus confirmed in favor of the experimental group.

### 3.5. Group Differences

The two groups differed in several key characteristics. The experimental group consisted of patients diagnosed with both depression and alcohol dependence, with a tendency toward higher extroversion. Their therapy sessions were scheduled early in the morning, serving as the first activity of the day in the clinic. In contrast, the control group primarily included patients with severe depression and minimal comorbid psychological disorders. Their therapy sessions took place later in the day, starting after the lunch break.

## 4. Discussion

### 4.1. Self-Efficacy

It turns out that the EG was more depressed than the CG at the start of the study and had lower values for the LK and FS. Only the self-efficacy value of the EG was higher than in the CG. One explanation for this could be the fact that the EG consisted of patients who, among other things, suffered from alcohol dependence, and some participants had only been abstinent for a short time. Many former alcohol addicts suffer from anxiety and depression, especially in the first few months of abstinence, which is due to the phase of neurobiological adaptation to the new “normal state” without alcohol ([2]). In the first few weeks, the brain is particularly vulnerable due to changes in neurotransmitter systems, and support measures in this phase are crucial ([13]). Abstinence in the clinic was mandatory and was regularly tested with urine samples. Since most of the EG patients were able to maintain abstinence, which is not easy, this could serve as an explanation for the EG’s increased self-efficacy values. In addition, 3 months later, both groups showed values approaching the initial value again. The reason for this could be the end of the intervention and, in the experimental group, the waning effects of abstinence. It is possible that the feeling of self-efficacy is stronger at the beginning of abstinence, as abstinence is initially perceived as an extremely difficult task to be mastered, and some patients were becoming abstinent for the first time in a long time in this study.

### 4.2. Depression

In addition, this could also serve as a reason for the higher depression scores and lower art of living and flourishing scores in this group, since the EG group had to struggle with alcohol abstinence and was therefore more susceptible to negative emotions during this phase. The patients in the EG showed an extreme drop in depression scores after 4 weeks. In comparison, the patients in the CG showed only very slight but not significant changes. This difference seems so extreme that it cannot be explained by the intervention alone. In part, the intervention can be used as a reason, but on the other hand, extreme differences were also shown in the group composition. The EG, which consisted mainly of alcohol addicts, was, on average, much more open and showed close bonds with one another ([28]). The CG consisted of a large number of introverted patients who hardly interacted with each other during the intervention or clinic breaks. In comparison, the initial study also showed a very extreme effect in the experimental group when using the art of living intervention. In this experiment, the strongest effects were seen in the depression variable too. As described in the theoretical background, it is clear that group therapy can bring about a significant improvement in depressive symptoms, especially in depressed patients.

### 4.3. Flourishing

The flourishing values in the CG only increased minimally and fell below the initial value after 3 months. The reason for this could be a short-term deterioration in well-being, which is often found in patients after the end of therapy ([3]). In comparison, the values in the EG increased sharply after 4 weeks and only fell slightly after 3 months. Here, the group effects (alcoholics, openness) can be used as an explanation again. In another study ([28]), there was also a drop in values in both groups after 3 months.

### 4.4. Art of Living

Only with the art of living scores was there a steady improvement in the EG even after 3 months. In the EG, there was no change after 4 weeks and a decline after 3 months. In the EG, one possible reason for the steady improvement is that the intervention is specifically aimed at the art of living. Some of the patients reported that they had benefited from the intervention even after the end of the therapy and automatically applied it again and again.

### 4.5. Group Effects

In addition to the intervention, the group effects were significant. The clinic divided up the two groups in advance with the aim of bringing similar people together. In particular, the alcohol-dependent patients were grouped together as one group. Judging by behavioral observations, the CG, consisting of introverted patients, showed little openness and preferred to spend breaks quietly and alone. Little contact was observed between them. Since it was a continuously changing group, the group composition was constantly mixed. In contrast to closed groups, which exist with a fixed number of participants over a set period of time, an ongoing group remains dynamic: the composition of the group changes regularly as people join and leave. This type of group offers flexibility and allows new patients to start therapy immediately without having to wait for a new group to start. However, it also generates challenges, such as the need to integrate new members into the existing group dynamic, which must be taken into account when assessing therapy outcomes and the stability of the group. This did not seem to be a problem in the EG, as the patients quickly found their way into the group, and the openness meant that the group quickly grew together. It was different in the EG. In this group, every new patient seemed to increase the closedness of the group. However, the ongoing group mainly increased the external validity of the results. The patients in the EG had poorer starting values in almost all the variables compared to the CG, which could be explained by the clinic’s group division into alcohol-dependent patients vs. non-alcohol-dependent patients. Another difference was the timing of the group therapy. The CG always had group therapy late in the afternoon at the end of the therapy day. At that time, the patients always seemed less receptive and unmotivated. It is possible that the majority of the patients had a midday slump and that the previous therapy day had already required a lot of cognitive effort. The EG always had therapy in the morning, at the start of the therapy day. It is possible that they had more motivation and better cognitive performance at the beginning of the day. In addition, the intervention was easier to carry out because it included questions about the previous day, and patients were probably able to recall the previous day better early in the morning than in the late afternoon. In addition, the day could be started directly with positivity and the effects of the intervention. In both studies, improvements in the values of the LK, FS, depression and SWE scores were shown in the groups with the art of living intervention. In an individual setting, the internal and external validity is increased. The group study showed group effects as well as selection effects (alcoholics) and spillover effects due to the motivated behavior of the experimental group and the reserved and unmotivated behavior of the control group. These effects could be responsible for the results, among other things. Nevertheless, a very strong effect was shown in the group with the art of living intervention, which suggests the need for further research in this area.

Further limitations, particularly for experimental studies like this one, include limited scopes, where a study focuses narrowly on specific variables or conditions, potentially overlooking broader contextual or interacting factors. Practical constraints, such as resource limitations, time restrictions, or logistical challenges, can restrict the scale or depth of a study. Randomization challenges make it difficult to achieve truly random assignment of participants to groups, which is essential for minimizing selection bias. Additionally, the presence of confounding variables—factors outside the control of the experiment that may influence the results—can complicate the interpretation of findings, making it difficult to establish a clear and definitive causal link between the variables under investigation. Future research could prioritize longitudinal studies conducted over an extended timeframe to gain deeper insights into the progression of changes and to detect significant differences between groups. By tracking participants over a longer period, such studies could capture more nuanced patterns of improvement, stability, or decline, providing a more comprehensive understanding of the long-term effects of interventions.

## 5. Conclusions

Overall, it can be seen that, similar to the initial study, the EG showed a greater improvement in all variables after 4 weeks than the CG. In the follow-up measurement, only the variables LK and depression were significantly improved in the EG.

At the beginning of the study, the EG, which consisted mostly of alcohol-dependent patients, showed better values for self-efficacy but worse values for the art of living, flourishing and depression compared to the CG. The EG had to struggle more with depression and negative emotions due to abstinence. The EG’s self-efficacy was initially high because successful abstinence was perceived as a difficult task, and it fell slightly after three months because the effects of abstinence wore off.

A sharp decline in the values for depression was visible in the EG, while hardly any changes occurred in the CG. This was possibly partly due to the effect of the intervention as well as group effects: the EG was more open and more connected, while the CG, consisting of more withdrawn patients, interacted little. Group effects could also have contributed to the stronger increase in flourishing values in the EG, while those in the CG rose minimally and then fell again. The EG also showed a steady improvement in the art of living over three months, while the CG showed only minimal effects here.

The different group compositions, the continuous renewal of the groups and the different therapy times (EG in the morning, CG in the afternoon) could have influenced the results. The selection effects and spillover effects caused by the group dynamics influenced the internal validity of the results. The study in the individual setting showed significant main effects, which were weakened in the group study by group effects and selection effects. In future studies, further, longer-term group effects could be investigated, and the intervention could be linked to personality types. It is possible that people who are more open benefit more from the intervention. It is also possible that in a suitable group composition with, for example, motivated patients, stronger effects can be generated by the intervention. Furthermore, one could investigate the extent to which certain mental disorders correlate with the intervention and the results and thus, for example, whether patients of a certain disorder group are more suitable for the intervention. Overall, the findings of this study indicate that group therapy has a positive impact on depression scores. However, this does not diminish the value of individual therapy. Both group and individual therapies offer distinct advantages and limitations, and it is important to assess each patient’s unique needs and circumstances. A personalized approach, considering the specific characteristics and preferences of each patient, is recommended to determine which form of therapy would be most effective for them.

The findings of this study have valuable implications for clinical practice. They highlight the importance of carefully considering group composition to ensure that participants are well matched in ways that may enhance therapeutic outcomes. Additionally, the study underscores the significance of scheduling, suggesting that paying closer attention to the timing of therapy sessions could impact their effectiveness. Furthermore, the results suggest that allocating more clinic time to group therapy sessions could be beneficial, as group-based interventions appear to play a crucial role in patient progress and recovery.

## Figures and Tables

**Figure 1 behavsci-15-00057-f001:**
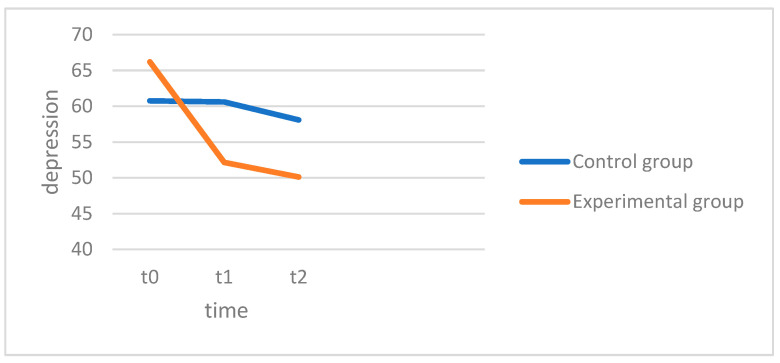
Depression.

**Figure 2 behavsci-15-00057-f002:**
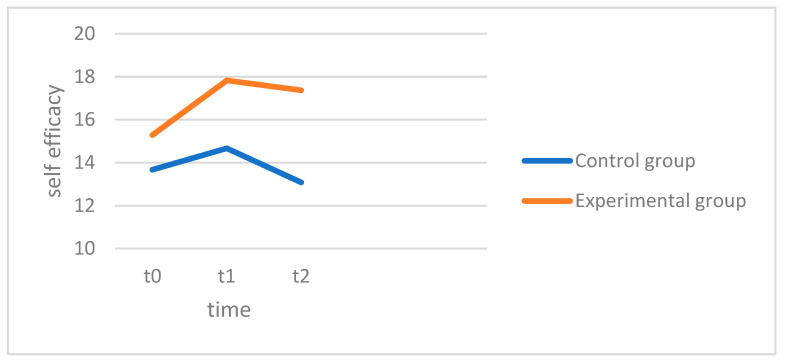
Self-efficacy.

**Figure 3 behavsci-15-00057-f003:**
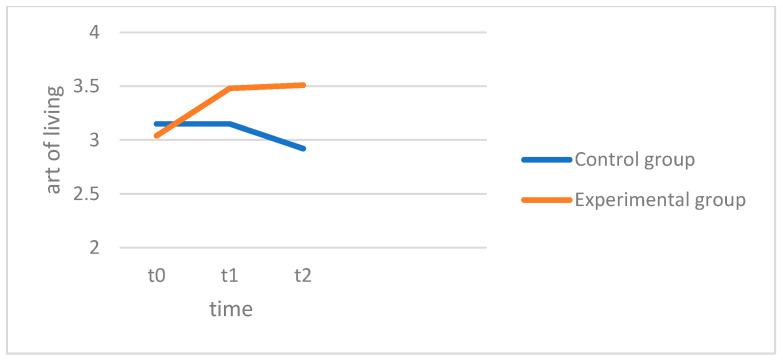
Art of living.

**Figure 4 behavsci-15-00057-f004:**
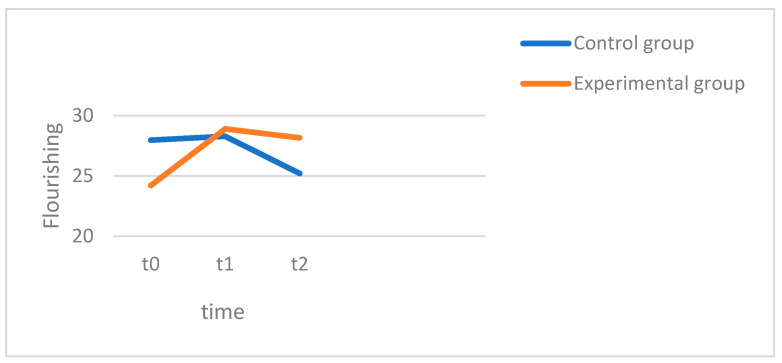
Flourishing.

## Data Availability

Data are available in a public repository; the data presented in this study are openly available in Zenodo at https://doi.org/10.5281/zenodo.14204749, accessed on 22 November 2024.

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
