# Peer review of "Shared Paths to Well-Being: The Impact of Group Therapy"

_behavsci, 2025, doi:10.3390/bs15010057_

Round 1

Reviewer 1 Report

Comments and Suggestions for Authors

Learning Happiness Together: The Impact of Group Therapy on Boosting Well-Being

First of all, thank you very much for allowing me to collaborate in the revision of this article, in order to provide another view to researchers and to increase its quality. 

The following is a structured analysis of the manuscript:

Attending to the TITLE, it is observed that it has clarity, reflects the content and the main focus of the study. 

Regarding the ABSTRACT, it includes the objectives, design, findings and relevance of the study and points out the key constructs (depression, life satisfaction, self-efficacy and ‘art of living’). However, the authors are advised to include the limitations of the study, ‘those typical of experimental studies’, and quantitative results such as ‘the experimental group showed a 25% reduction in depression scores and a 20% increase in life satisfaction’.

Regarding the INTRODUCTION, the theoretical framework provided by the authors is adequate, it highlights the prevalence of mental illness and the effectiveness of group therapy, in addition, it cites relevant literature on group interventions and psychological constructs, but some references are prior to 2018 and considering that this is a field in which there are current and abundant findings, it is recommended that the authors update the citations on cognitive group therapy for depression. Likewise, it is noted that although the theoretical framework is well constructed, the inclusion of the construct ‘art of living’ in the conceptual framework is not sufficiently justified and should be done. 

As for the MATERIALS AND METHODS section, the experimental design is well structured, with control and experimental groups and measurements at three time points, and the exclusion criteria and measures used (BDI-V, SWLS, Flourishing Scale) are detailed. However, the group assignment is not clearly explained and information on the validation of the questionnaires in the specific population is missing, it is necessary to specify the randomisation and to include a description on the adaptation of the scales to the cultural context, indicating whether the tools were translated and validated in the German population in previous studies, and indicating, if possible, the reference to any of these studies.

Regarding the RESULTS, the section presents clear and significant correlations between constructs and the findings are well organised by hypotheses and constructs (depression, self-efficacy, life satisfaction, ‘art of living’). However, it is recommended that the authors: a) incorporate clear graphs for each key variable, adding information about it in the section (a graph is already included, but detailed interpretation is missing in the text) and b) explain group differences, related to baseline characteristics (e.g., alcohol dependence).

With regard to DISCUSSION, the section connects the results obtained with previous research noted in the theoretical framework, highlighting the effectiveness of group therapy and indicating additional factors such as group dynamics and intervention schedules. However, it is recommended that the authors of the paper acknowledge methodological limitations and point out future lines of intervention, such as longitudinal studies, for example.

Regarding the CONCLUSIONS, they highlight the positive effects of the experimental intervention on depression and ‘art of living’. But the authors are recommended to propose practical applications and to determine applicability in other clinical contexts.

In view of these issues, it is proposed to the editor to ACCEPT THE ARTICLE FOR PUBLICATION, ON ACCEPTANCE OF THESE SMALL MODIFICATIONS BY THE AUTHORS.

Author Response

Thank you very much for your thorough revisions. I edited them all and attached the file "cover letter" with all revisions made.

Reviewer 2 Report

Comments and Suggestions for Authors

This research clearly shows the benefits of Group Therapy in the chosen groups to treat depression. However, my main concern is that the article is based heavily on situations where alcohol and substance dependence interfere with therapy and there obviously are marked differences between the participants and the ways how they feel about participating in group therapy (as the drop-out rates indicate). Learning happiness together works for some, but not for all – and there is a major risk that this study can be interpreted as a justification to use group psychotherapy as a cost-cutting method while it surely does not remove the need for other therapy methods. Maybe a bit more balanced language, particularly in the title and conclusions, would be in place. 

Author Response

(The authors gave the same response as above.)
